# Micro RNAs—The Small Big Players in Hepatitis E Virus Infection: A Comprehensive Review

**DOI:** 10.3390/biom12111543

**Published:** 2022-10-22

**Authors:** Elitsa Golkocheva-Markova

**Affiliations:** NRL Hepatitis Viruses, Virology Department, National Center of Infectious and Parasitic Diseases, 1233 Sofia, Bulgaria; elmarkova2007@gmail.com

**Keywords:** HEV, HEV-miRNA, miRNA-122, miRNA-214

## Abstract

The molecular mechanism of hepatitis E virus (HEV) pathology is still unclear. The micro RNAs (miRNAs), of host or viral origin, interfere with virus replication and host environment in order to create an appropriate condition for the production of mature HEV progeny. Understanding the biogenesis and the interference of miRNAs with HEV will help to revile the mechanism of viral pathogenesis.

## 1. Introduction

Micro RNAs (miRNAs) are a class of small non-coding RNAs that are conserved in eukaryotes and are essential in the control of translation and transcription rate. In humans, they are crucial for cell differentiation during embryogenesis, for cell–cell communications and for immune response to infections. At the same time, they interact with different pathological processes as cancer development or a generate environment suitable for pathogens’ replication. In recent years, evidence for the role of miRNAs in viral replication and adaptation has accumulated. The first viral miRNA was detected in cell culture system during latent Epstein–Barr virus infection [1]; since then, 569 viral miRNA sequences have been deposited in the microRNA database (miRbase) as the majority of them are generated by DNA viruses [2], Despite consistent limitations, small viral genome and replication in the cytoplasm, RNA viruses also encode miRNAs (rv-miRNAs).

Hepatitis E virus (HEV) is transmitted mainly by the fecal–oral route and is the causative agent of acute viral hepatitis worldwide. HEV is classified into eight genotypes [3], as HEV genotypes 1 (HEV-1) and 2 (HEV-2) infect only humans and are endemic in low-income countries [4]. Acute infection with HEV-1 and HEV-2 in pregnant women can progress to fulminant liver failure [5]. Genotypes 3 (HEV-3) and 4 (HEV-4) are zoonotic and are autochthonous, with sporadic cases in high-income countries. In most cases, infection caused by zoonotic genotypes is mild and self-limited, affecting mainly middle-aged and elderly males, but in immunocompromised patients, HEV-3 can cause chronic infection [6].

HEV is a positive sense single-strand RNA virus, in which the virion presents in two forms: enveloped and non-enveloped. The viral genome contains three open reading frames (*ORF*)). From all genotypes, which infect humans, HEV-3 is characterized by high variability [7], and recombination usually occurs in different regions of *ORF1* [8]. HEV replication is a complex process with production of different RNAs replicative intermediates, which may imply the existence of rv-miRNA.

The aim of the present comprehensive review is to provide a brief overview of the host and predicted viral miRNAs with an essential role in HEV pathogenesis.

## 2. Hepatitis E Virus

### 2.1. Taxonomy and Etiology

According to the new Master Species List #37 of the International Committee on Taxonomy of Viruses (ICTV), HEV belongs to species *Paslahepevirus balayani*, genus *Paslahepevirus*, family *Hepeviridae* in the realm *Riboviria* [9]. Recently, HEV was classified into eight genotypes [3,10]: genotypes 1, 2, 3, 4 and 7 infect humans. Different genotypes are characterized by different geographical distributions, routes of transmissions, pathogenesis and host ranges. HEV genotype 1 and 2 are exclusively infect humans, while genotype 3 and 4 are zoonotic and circulate among animal species (genotype 3: pig, wild boar, deer, mongoose, and rabbit; genotype 4: mainly pig) and humans; genotype 7 circulates in camels, but a case report documented transmission to a human [11]. In European countries, in most cases, autochthonous HEV infection is mild and self-limited, affecting mainly middle-aged and elderly males and is caused by HEV-3. Chronic infection is reported for immunocompromised patients, such as transplant patients, hematological patients on chemotherapy, HIV infected patients and those under treatment with corticosteroids and immunosuppressive agents [6]. Until recently, it was believed that chronic infections were caused exclusively by HEV-3, but some case reports suggest that prolonged persistence of genotypes 1, 4 and 7 in immunosuppressed patients can also occur [11,12,13]. HEV can cause fulminant hepatitis in patients with underlying liver disease, with mortality rate up to 30% in pregnant women infected with HEV-1 in the third trimester [5].

### 2.2. Genome Organization

Hepatitis E virus is a single-stranded RNA quasi-enveloped virus that in bile and feces is non-enveloped (neHEV) and in blood circulates as an enveloped form (eHEV). Its genome consists of three open reading frames – *ORF1*, *ORF2* and *ORF3*, flanked with untranslated regions (UTR). For HEV-1 the transient translation of *ORF4*, during endoplasmic reticulum stress, has been identified [14]. 5’-UTR is characterized by the presence of 7-methylguanosine cap structure (7mG), and 3’-UTR, by the polyadenylated tail (polyA) [15]. The entry of HEV into host cells is due to clathrin- and dynamin-dependent pathways, following by uncoating, depending on the membrane associated virus form, and delivery to cytosol. After entry, *ORF1*, which is the largest one, with structural domains for methyltransferase (*Met*), Y domain (*Y*), papain-like cysteine protease (*PCP*), hypervariable region (*HVR*), X domain (*X*), helicase (*Hel*), and the RNA-dependent RNA polymerase (*RdRp*), is translated by the cap-dependent mechanism directly from the positive HEV RNA into a polyprotein [16]. Serving as a template, the positive sense strand is transcribed by viral RdRp to negative sense full length replicative intermediate, which serves as a template for the transcription of the full-length positive sense RNA molecule and smaller positive sense subgenomic bicistronic mRNA. The first is packed into progeny virions, and the second one encodes *ORF2* and *ORF3* [16,17]. *ORF2* encodes viral capsid proteins. *ORF3*, which is overlapping *ORF2*, encodes a small phospoprotein that facilitates the virions release in the quasi-enveloped form [16]. ORF1 polyprotein, by the involvement of different domains, inhibits IFN-β (Type I IFN) expression and ferritin secretion (involved in iron transport) [18], and increases virus replication by interaction with host miRNAs [19]. At the same time, the structure of the HEV genome is very similar to the host mRNA, that helps the virus to escape detection by immune sensor molecules [20]. The underlying mechanisms are varied and happen via the ability of HEV to act during replication on the different steps of the IFN induction [21]. In fact, the HEV proteins interact with the host immune system in order to facilitate virus entry and replication. It is worth mentioning that *ORF1* is the most susceptible to recombinant events, insertions and deletions, as the last are typical for *HVR*. Examples for insertions, with host origin, are the Kernow-C1 HEV-3 strain that contains 171-nucleotide segment of the S17 human ribosomal protein gene, and HEV strain containing fragments from the human S19 ribosomal protein gene, and from the human *TAT* and *ITI-H2* genes [22]. All strains were isolated from chronically HEV infected immunocompromised patients and are characterized with enhanced HEV replication in the cell culture model. The single nucleotide polymorphism events predominantly happen in the *RdRp* domain and are connected with treatment failure as virus adaptation to the changing environment [8]. Recombination events are typical for *Hel* and *X* domains of *ORF1* [23].

### 2.3. Tropism

HEV is a primary hepatotropic virus, but hepatitis E virus infection can also proceed with extrahepatic manifestations, such as neurological and renal diseases, hematological disorders, pancreatitis, myocarditis, arthritis and autoimmune thyroiditis [24]. In the animal experimental model, HEV-3 has been isolated from various tissues and organs, lymphoid organs, gastrointestinal tract, kidneys and from the peripheral nerves. The heart is the only organ where HEV Ag was not detected [25]. Additionally, HEV Ag was detected in the kidney of gerbils, experimentally infected with swine HEV [26]. HEV RNA was detected in the kidneys of different experimentally infected animals [27]. The kidney, as a possible site for HEV replication in humans, was confirmed by the detection of HEV RNA in the urine of a chronically HEV infected patient [28].

## 3. miRNA and the Viral Hepatitis

miRNAs are small (up to 25 nt), non-coding highly conserved RNAs, which represent 1% of the human genome but interact with about 60% of messenger RNAs [29]. The matured mi-RNAs recognize their complementary mRNAs through the construction of the miRNA induced silencing complex. The base-pairing occurs between the so-called “seed region” (located on 2nd to 7th nt) within miRNA’s 5’-end and between the miRNA binding site, located within 3’-non-translated region (NTR) from the host mRNA or the virus genome. For RNA viruses, these miRNA binding sites can be located in the 5’-, 3’- NTR or in the coding regions of viral proteins [30]. When the base pairing is incomplete, the partial duplex between complimentary sites is formed, thus leading to translational inhibition and mRNA destabilization and in some case degradation [31]. The host miRNAs bind directly to the genome of the positive stand RNA viruses by acting through two different mechanisms: 1) inhibition of viral translation; and 2) stabilization of the viral RNA and enhancing replication [30]. The miRNA expression signature is tissue- and disease- specific [32]. In the liver, the main pool is presented by approximately 10 different mi-RNAs with a predominance of miRNA-122, which is the most highly expressed miRNA in the adult human liver [33] and presents around 70% of the mi-RNA pool. This mi-RNA is involved in the proliferation and differentiation of the liver cells [34] and regulation of the fatty acid metabolism in the liver [35]. In the mice infection model, inhibition of miRNA-122 leads to the reduced expression of lipogenic genes [36]. Acute infection with hepatotropic viruses is characterized by inflammatory-cell infiltration and hepatocellular damage [37]. Thus, the virus infection alters the mi-RNAs liver-specific pattern.

Viral hepatitis—hepatitis A virus (HAV), hepatitis B virus (HBV), hepatitis D virus (HDV), hepatitis C virus (HCV) and HEV— are hepatotropic viruses belonging to different families: *Picornaviridae*, *Hepadnaviridae*, *Kolmioviridae*, *Flaviviridae*, and *Hepeviridae* [38]. HAV, HDV, HCV and HEV are RNA viruses, as HBV is a DNA virus. In addition, HDV is a satellite virus, and its transmission and release depend on HBV.

Hepatitis C virus is a positive sense RNA virus, whose genome contains one ORF, flanked by 5’ and 3’ UTRs. Within 5’-UTR, the internal ribosome entry site is located, which is essential for cap-independent translation of the HCV genome [39]. In 5’-UTR are located two miRNA binding sites for host miR-122, which are essential for efficient virus replication [40]. It is supposed that miR-122 binding suppresses alternative folds of the HCV RNA 5′ UTR, leading to the formation of a canonical IRES structure, and the stabilization of the HCV genome is stimulated [41]. Thereby, miR-122 enhances HCV replication. The positive effect of the miRNA-122 on HCV replication has been established, as the microRNA complimentarily binds to two separate miRNA binding sequences located within the first stem loop at 5’-UTR of the HCV RNA and thus induces the translationally active IRES (internal ribosomal entry site) structure [42,43]. It was established that other miRNAs—miR-448, miR-196, miR-199a, let-7b and miR-181c—interact directly with HCV, but they inhibit virus replication [29]. It can be hypothesized that, together with decreased viral replication, the interaction between HCV and liver mi-RNAs, involved in lipid metabolism, is essential in the establishment of persistent HCV infection [44]. It is supposed that HCV does not encode viral miRNAs [34].

HBV is a small DNA virus, with partially double-stranded relaxed circular DNA, comprised of four overlapping ORFs. The HBV DNA is replicated by reverse transcription and the formation of pregenomic RNA. Cellular miRNAs act directly with HBV transcripts or indirectly by targeting cellular factors relevant to the HBV life cycle. In most cases, direct acting mi-RNAs—miR-122, mir-125 family, miR-199a-3p, and miR-210—inhibit HBV replication, which is supposed to be associated with the establishment of persistent HBV infection [45]. At the same time HBV modulates host miRNA biogenesis by interaction with polymerase complex, essential for mature miRNA synthesis [45]. HBV encodes two viral miRNAs (HVB-miR-2, -3) that target the viral transcript, leading to a decrease in HBV replication and acting as oncogenes. It is supposed that this is the mechanism by which HBV leads to mild liver pathogenesis and the establishment of chronic HBV infection [46].

Till now, the possible encoding of viral miRNAs by HDV has not been well studied. The HDV genome is a single-stranded negative-sense covalently closed circular RNA molecule with approximately 74% base pairing. It is hypothesized that HDV is a result of the recombination between a viroid-like element and a host mRNA [47]. At the same time, HDV RNA and antigenome RNA are resistant to enzymes acting in the miRNA-induced silencing complex [48].

HAV is a typical cytoplasmic virus that is nonlytic and is released as an enveloped form in the patient’s serum and cell supernatant and as a non-enveloped form in feces [49]. In fact, despite the notion that cytoplasmically localized RNA viruses do not encode microRNAs, HAV is the first in which such expression has been demonstrated [50]. It is supposed that HAV encoded three viral miRNAs (hav-miR-1-5p, hav-miR-2-5p and hav-miR-N1-3p) which decrease dramatically HAV RNA synthesis in the cell culture model helping in the establishment of a non-cytopathic persistent infection [51]. The same role was established for 94 host miRNAs expressed during the HAV fibroblast infectious model, which attenuates HEV replication by affecting the type I interferon signaling pathway [52].

As can be summarized, the interplay between hepatitis viruses and viral or host miRNAs can be direct, by complimentary binding between miRNA and viral target sequence, or indirect, by alteration of the host signal pathway during virus invasion and replication. In the case of HCV, the liver miRNAs are essential for viral genome stabilization and the establishment of persistent infection with maintenance of low viral replication. For HBV and HAV, these interactions are focused on persistent infection. Thus, it can be hypothesized that hepatitis virus liver tropism appears to be due to the positive effect of liver miRNAs.

## 4. Hepatitis E Virus miRNA

In addition to viral transcripts, HEV produced several different miRNAs. Till now, their study has mainly been based on prediction in silico models confirmed by in vitro or in vivo experimental infectious models. At present, these studies are hampered by the absence of a suitable infectious model, as HEV replicates slowly in cell-cultured systems, leading to virus adaptation [53].

### 4.1. HEV-miR-A26

The coding sequence for HEV-miR-A26 was located in the *MeT* domain (Figure 1) from HEV *ORF1*, and the sequence is conserved among different genotypes.

The production of HEV-miR-A26 by HEV was confirmed in the cell transfection model, animal infection model and measurement of increasing miR-A26 levels in patient’s serum from an acute phase, vs. patients in convalescence phase of HEV infection. It is hypothesized that the main function of HEVmiR-A26 is to facilitate virus replication by suppressing the production of IFN-β [54].

### 4.2. HEV-miRNAs

Using computational prediction modeling, nine potential HEV-miRNAs—HEV-MD1, HEV-MD2, HEV-MD3, HEV-MD25, HEV-MD31, HEV-MD-35, HEV-MD39, HEV-MR9 and HEV-MR10—were identified for HEV-1 (Figure 1). Positions of mature mi-RNAs coding sequences are located within different domains of the *ORF1* (*Met, PCP* and *RdRp*) and within *ORF2*. HEV-miRNAs target sites are proposed to be located at both 3’-end and 5’-end of human mRNA. The targeted silencing has been proposed for genes, essential in chromosome organization, cell differentiation, lipid and nitrogen metabolism, membrane organization and transmembrane transport, and cell–cell signaling. From all nine predicted HEV-miRNAs, five—HEV-MD1, HEV-MD2, HEV-MD25, HEV-MD31 and HEV-MR10—affect membrane organization and transmembrane transport [55]. As was mentioned, HEV circulates in an enveloped form in the supernatant of the infected cell culture and in the serum from acute HEV-positive patients [15], which can explain the virus action to facilitate the progeny envelopment and further the dissemination within the host. It was predicted that HEV-MD2 targets the production of cyclin G-associated kinase [45], which is essential for clathrin trafficking, by mediating binding of clathrin to the plasma membrane, and regulating receptor signaling by influencing trafficking downstream of clathrin-coated vesicles [56]. The membrane internalization of non-enveloped HEV is clathin- and dynamin dependent [57], which suggests that such targeting is beneficial for HEV membrane trafficking. Thus, one of the putative actions of the predicted HEV-miRNAs is to facilitate the virus transport inside and outside the host cell and virus egress, as the possible extrahepatic HEV dissemination. However, the expression and mode of action of annotated HEV-miRNA should be confirmed by experimental approaches.

## 5. Host miRNA Affecting HEV Life Cycle

Prediction models for human host miRNAs, targeting HEV genome, are based on previously studied miRNA signatures in viral hepatitis infection, such as HCV and HBV, and on host miRNA cell-, organ-, and tissue-specific patterns. To date, the most extensively studied host miRNAs affecting HEV replication are miR-122 and miR-214.

### 5.1. miRNA-122

In humans, the gene coding miRNA-122 (miR-122) is located within chromosome 18. As it was described above, miRNA-122 is the most represented in the human liver, and its role was extensively explored in liver cell differentiation, in cholesterol metabolism, as in liver pathogenesis, due to HCV- and HBV-induced hepatocellular carcinoma. 

During the interplay between miRNA-122 and HEV, the viral replication is affected [19]. The direct complimentary binding of the miRNA-122 and miRNA binding site, usually located in the *RdRp* region from the *ORF1* gene (Figure 1), enhances HEV replication. The number of predicted binding sites within the HEV genome are genotype specific. In fact, the HEV genotype 1, which infects only humans, possesses one conserved miRNA biding site, located in the *RdRp* region of *ORF1*. In some strains, the conserved binding site presents in combination with one or two additional sites, located in the *ORF1* or *ORF2* regions of a virus genome (Figure 1). It can be hypothesized that additional binding sites have a “rescue” function in the case of mutation. Thus, for HEV-1, a relatively conserved distribution of miRNA binding sites was established. This conservation can be assumed to be the result of viral evolution in contexts of the adaptation to a specific host environment; in the case of HEV-1, this is the human. In zoonotic HEV-3 strains, isolated from humans, the miRNA-122 binding sites are distributed at different nucleotide positions within *ORF1* without any conserved pattern. For HEV-3 strains, isolated from pigs, the target sites are located within variable genomic regions [19]. At the same time, HEV-3 strains, of human or animal origin, cannot be differentiated by the distribution of miRNA binding sites, the action of the same miRNA in both hosts, or a process of current viral evolution with respect to the host. The direct interplay between miR-122 and the HEV genome was confirmed by immunoprecipitation and molecular approaches [19]. This direct interaction resulted in decreased serum levels of circulated miR-122 from patients with acute HEV infection in comparison with non-infected controls [58], or in acute viremic patients vs. chronic viremic and non-viremic ones [59]. Thereby, comparative analysis of circulating miRNA levels among HEV acute patients and healthy controls can be used as a predictive pattern for active HEV replication.

In recent years, the pivotal role of miR-122 in regulation of hepatocyte innate immunity was established, and was achieved by the repression of negative IFN regulation [60]. Thus, it can be hypothesized that by modulating miRNA-122 levels, as a direct complimentary binding, HEV affects the host inflammatory response and induces liver inflammation during infection [58]. The same correlation was detected for chronic hepatitis B patients, where the decrease in miR-122 levels led to enhanced inflammatory cytokine (C-C motif chemokine ligand and interleukin-6) expression [61]. 

### 5.2. miRNA-214

The gene coding host miRNA-214 (miR-214) is located in a chromosomal region 1q24.3, in intron 14 of the Dynamin-3 gene (DNM3) [62]. MiR-214 exerts morphogenesis of the muscles, skeleton and nervous system, and at the same time plays a diverse role in the development of different cancers types; it stimulates the progression of gastric, ovarian and lung cancer, but suppresses liver and cervical cancers [63]. HEV is a hepatotropic virus with possible link for development of hepatocellular carcinoma after chronicity in immunocompromised patients or in presence of underlying liver disease [64]. Thus, looking for binding sites for miRNAs with an established role in liver pathogenesis is understandable. 

Through the comparison of the HEV whole genome sequences, the possible conserved miRNA-214 binding sites were established for HEV genotypes 1, 2, 3 and 4, and these sites are located within *PCP*, *HVR*, *Hel* and *RdRp* domains at *ORF1* (Figure 1). Conservation was established for both human (genotypes 1 and 2) and zoonotic (genotype 3 and 4) HEV strains, despite the high mutation rate of HEV: 1.5 base substitutions per site per year [23]. In this way, the unusual maintenance can be explained by the miR-214 beneficial effect on HEV life cycle and the ongoing viral evolution for its preservation. miR-214 enhanced HEV replication by different mechanisms: 1) positive regulation due to direct binding to HEV RNA and enhancement of virus genome translation that leads to elevated production of structural ORF2 and ORF3 proteins; 2) synergy increasing in miR-214 levels; and 3) negative regulation of a protein C levels and increased thrombin activity [65]. It was hypothesized that intracellular thrombin, in synergy with factor Xa, mediates the cleavage of the ORF1 polyprotein during HEV replication, thereby releasing the viral RdRp, which is essential for the synthesis of genomic (positive sense) and anti-genomic (negative sense) strand, and for the replication of HEV *ORF2* and *ORF3* and synthesis of the viral structural proteins [66]. The role of thrombin in the blood–brain barrier damages is well defined [67], which, in combination with enhanced thrombin activity during active HEV replication, can be exploited as an explanation for the development of acute neurological symptoms.

## 6. miRNA Pattern during HEV Infection

The main function of miRNA in host defense is an alteration of the viral replication (1) by directly binding to the viral genome; (2) indirectly, by stimulating the secretion of pro-inflammatory cytokines; or (3) by stimulating changes in the miRNA pattern at the cell or tissue level [30].

The miRNA-specific signature, that is characterized with a specific miRNA pool, was described for different human tissues [32]. For example, miRNA-122 is the most highly expressed miRNA in the adult human liver [33] and presents around 70% of the mi-RNA pool in liver [34]. Additionally, miRNA-122 is presented in the miRNA signature of other tissues, as the lung, gallbladder, pancreas, kidney, spleen, stomach and nerves (Figure 2).

The miRNAs cannot be detected only in tissues or specific cells; they circulate in the serum and other different extracellular bio-fluids, such as cerebrospinal fluid, saliva, tears, urine, peritoneal fluid. They are characterized by a typical expression profile in healthy individuals [68]. Changes in the miRNA pattern are indicative not only of the process of differentiation, but also of pathological events, such as viral invasion and replication. Studies on the miRNA pattern suggest the possible link between different host miRNAs and HEV. Changes in the levels of miR-628, miR-151, miR-526b, miR-1285, miR-520b, miR-302b, miR-627 and miR-365 were detected for HEV RNA positive patients and blood donors in comparison with negative blood donors [69]. A specific expression profile was revealed during acute self-limiting HEV-1 infection in pregnant women in comparison with HEV-infected non-pregnant women. The predominant expression of miR 188, 190b, 374c, 365a, 365b, 450a-1, 450a-2, 450b, 504, 580, 616, 2115, 3117, 4482, 4772 and 5690 was detected, as differentiation can be determined between acute infection, acute liver failure and self-limiting acute HEV infection [70]. The upregulation of miR-99a, miR-122, and miR-125b was measured in acute and chronic HEV-3 infected patients in comparison with healthy controls [59]. Specific miRNA signatures have also been found to differentiate acute and chronic HCV and HBV infections, or hepatocellular carcinoma as consequence of these infections.

## 7. Conclusions

As can be seen, HEV has adapted to host miRNA action in order to enhance replication and the production of mature progeny. HEV-miRNAs are acting mainly on later stages and facilitate the excretion and dissemination of the infectious progeny. In fact, the HEV-miRNAs coding sequences, as host miRNAs binding sites, are located predominantly in *ORF1*, which encodes non-structural viral proteins essential for viral replication. This is the most heterogenic/variable region from the HEV genome prompt to naturally occurring recombination, insertion or deletion. Zoonotic genotypes, HEV-3 and HEV-4, that infect humans and a broad range of animal species, are characterized with higher intra-species variability in comparison with HEV-1 and HEV-2 genotypes that infect only humans. Recombination usually occurs in the *Hel* domain of *ORF1*, as insertions, which can be of viral or host origin happen in the *HVR* domain [23]. Usually, all of these events facilitate virus replication or adaptation to the host environment and are the result of positive selection acting on the generated intra-host heterogeneous HEV mutant cloud. Despite the high variability of some domains, HEV acquires and maintains conserved binding sites for host miRNAs, which are beneficial for its replication. In fact, potential HEV-miRNA coding sequences have been studied in HEV-1 (restricted to humans) that are characterized with intra-species homogeneity. Thus, the prediction models, based on comparing sequencing data from different strains within the current genotype 1, are working considerably well. Expanding the computational prediction and experimental HEV-1 models to HEV zoonotic genotypes will help to evaluate the role of miRNAs in HEV adaptation, tissue and host tropism.

## Figures and Tables

**Figure 1 biomolecules-12-01543-f001:**
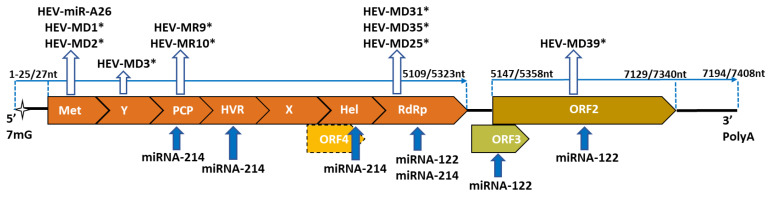
A schematic diagram of the target sites for host mi-RNAs and predicted encoded sequences for HEV-miRNAs. The HEV genome is approximately 7.2 kb and consists of three open reading frames (*ORFs*). *ORF1* is the biggest and encodes nonstructural proteins. *ORF2* is located ~30 nt. downstream from the stop codon of *ORF1* and encodes viral structural proteins. *ORF3*, which is encoded small phosphoprotein, is located ~20 nt downstream from the stop codon of *ORF1* and overlaps *ORF2*. The transient *ORF4* is located within *ORF1* and is represented by a dotted line [17]. Empty arrows represent sites encoding HEV-miRNAs, and full arrows, host miRNA binding sites. Asterisk (*) indicates in silico predicted HEV-miRNA coding domains. Indicated length in nucleotides (nt) is according to Tong H. et al. [23].

**Figure 2 biomolecules-12-01543-f002:**
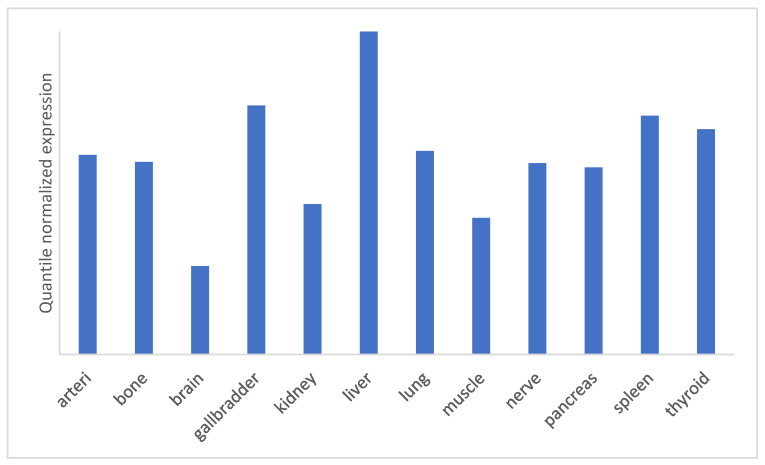
Expression plot of miR-122 in different organs. The expression of miRNA within different organs was calculated by tissue specificity index (TSI). Adapted from the tissue atlas [https://ccb-web.cs.uni-saarland.de/tissueatlas/ (accessed on 28 August 2022); [32]]. The maximum expression value for selected human organs is visualized. For best visualization, the quantile normalized expression is represented in the decimal logarithmic scale.

## Data Availability

Not applicable.

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
