# Peer review of "Micro RNAs—The Small Big Players in Hepatitis E Virus Infection: A Comprehensive Review"

_biomolecules, 2022, doi:10.3390/biom12111543_

Round 1

Reviewer 1 Report

In the present manuscript, E. Golkocheva-Markova provides a review on the interplay between hepatitis E virus and micro RNAs (miRNAs). A review on this topic is of great interest for the field as none (or very few) is available. Nevertheless, the present manuscript needs some improvements to be suitable for publication in Biomolecules. Some paragraphs are difficult to follow and details are sometimes missing to understand more clearly the results/conclusions gained from previous published studies on miRNAs and HEV. The paper needs additional English proofreading and some sentences need to be rephrased for clarity purpose.

Figure 1:

-          Are lines 183-189 belonging to the figure legend? It should be inserted after the figure title as the figure needs to be described in full details within the legends

-          Lines 183-184: already described in paragraph 2.2.

Figure 2:

-          Are lines 285-287 belonging to the figure legend? It should be inserted after the figure title.

-          More details need to be included about the data presented in the legend. How was expression calculated? The vertical axis title is missing.

-          Line 24/Ref 2: Indicate database website and date of accession.

 -          Line 39: ORF4 only found in HEV-1

 -          Line 58: common?

 -          Line 69: flanked rather than flanged

 -          Line 82: symbols for genes are italicized whereas symbols for proteins are not italicized

 -          In paragraph 3 on “miRNA and the viral hepatitis”, miRNA in general and the different types of hepatotropic viruses are listed and described individually without discussing specifically miRNA present in the liver or any similarities observed between viruses affecting the liver. It would be interesting to include such discussion.

 -          Line 138: HCV does not encode viral miRNAs or none has been identified yet?

 -          Line 68/157: faces instead of faeces/feces

 -          Line 196-203: this paragraph is very difficult to follow. Why is it suggested that HEV-miRNAs facilitate extrahepatic HEV dissemination and virus egress?

 -          Paragraph 5: this paragraph is very difficult to follow.

Author Response

I sincerely thank the reviewers for their valuable comments that led me to improve the manuscript. This manuscript was thoroughly amended in accordance with the reviewers’ recommendations and comments. I provided item-by-item responses to each point brought up and inserted them underneath each of them. Parts that are amended in the revised manuscript are highlighted in blue in a manuscript file with marked revisions.

Response to Reviewer 1 Comments

Figure 1:

-Are lines 183-189 belonging to the figure legend? It should be inserted after the figure title as the figure needs to be described in full details within the legends

Response: This was done as advised. The title of figure 1 has been formatted according to the journal's requirements, lines 183-189 have been inserted into the figure title, and a detailed description of the figure has been added.

-Lines 183-184: already described in paragraph 2.2.

Response: As recommended lines 183-184 were removed in order to avoid duplication.

Figure 2:

-Are lines 285-287 belonging to the figure legend? It should be inserted after the figure title.

-More details need to be included about the data presented in the legend. How was expression calculated? The vertical axis title is missing.

Response: This was amended as recommended. For figure 2, lines 285-287 have been inserted after the figure title, and the vertical axis title and the calculation method have been added.

-Line 24/Ref 2: Indicate database website and date of accession.

Response: I do agree with the reviewer that the database website and date of accession should be included, but this was cited from the work of Nanbo A, Furuyama W, Lin Z. RNA Virus-Encoded miRNAs: Current Insights and Future Challenges. 2021, Front Microbiol., Vol. 12, p. 679210. For this reason, the reference has been removed from the end of the sentence.

-Line 39: ORF4 only found in HEV-1

Response: Yes, the ORF4 is found only in genotype 1, so to clarify this line 39 has been removed and this has been described in paragraph 2.2 Genome organization, lines 71-72

-Line 58: common?

Response: Thank you for this remark, the word common has been removed.

-Line 69: flanked rather than flanged

Response: Thank you for this remark, on line 69 the spelling mistake has been corrected.

-Line 82: symbols for genes are italicized whereas symbols for proteins are not italicized

Response: This was amended as recommended, everywhere in the text the font of the characters for genes is corrected to italic

-In paragraph 3 on “miRNA and the viral hepatitis”, miRNA in general and the different types of hepatotropic viruses are listed and described individually without discussing specifically miRNA present in the liver or any similarities observed between viruses affecting the liver. It would be interesting to include such discussion.

Response: I do agree with the reviewer that more information about miRNAs in the liver will be interesting, but I think that their detailed description will go beyond the topic of the manuscript, for this more information on the liver miRNAs pool was added with attention to these miRNAs essential for HEV. In accordance with the reviewer’s comment in more details miRNA specific for different hepatitis viruses and their benefit are discussed (paragraph 3. miRNA and viral hepatitis)

-Line 138: HCV does not encode viral miRNAs or none has been identified yet?

Response: Yes, in the literature it is widely held that HCV does not encode viral miRNAs and none has been identified yet. The reference has been added to support this statement

-Line 68/157: faces instead of faeces/feces

Response: Thank you for this remark, on lines 68 and 157 the spelling mistakes were corrected.

-Line 196-203: this paragraph is very difficult to follow. Why is it suggested that HEV-miRNAs facilitate extrahepatic HEV dissemination and virus egress?

Response: I do agree with the reviewer's comments and the improvement of lines 196-203 was done. The discussion on the role of miRNAs in membrane trafficking of HEV was included, which has hypothesized their involvement in virus egress and extrahepatic dissemination.

-Paragraph 5: this paragraph is very difficult to follow.

Response: In accordance with the reviewer’s comment in paragraph 5 stylistic changes have been made to facilitate its understanding. Additional English proofreading has been done and some sentences are rephrased for clarity purposes.

Reviewer 2 Report

The manuscript entitled “Micro RNAs – the small big players in hepatitis E virus infection: a comprehensive review” by a present and comprehensive review of miRNAs, from both host and HEV, in the pathogenesis of HEV infection. Overall, it is a review that concludes our current knowledge regarding the miRNA-mediated regulation of HEV.   

Critics:

A figure depicting the mechanisms of how miRNAs are involved in the viral life cycle and host immunity will be beneficial for the readers.

Language can be further improved. For example, title: …, a comprehensive review; abstract: The molecular mechanism of the hepatitis E virus (HEV) pathology still is unclear. MicroRNAs (miRNAs), with the host or viral origin, interfere with virus replication and the host environment to create a suitable condition for producing mature HEV progeny. Understanding the biogenesis and interference of miRNAs with HEV will help reveal the mechanism of viral pathogenesis.

Overlaps have been found in contexts between 1. Introduction and 2. Hepatitis E virus; it could be more concise.

Errors: 4.1. HEV-miR-A(2)6; 5.1(2). miRNA-214

Author Response

I sincerely thank the reviewers for their valuable comments that led me to improve the manuscript. This manuscript was thoroughly amended in accordance with the reviewers’ recommendations and comments. I provided item-by-item responses to each point brought up and inserted them underneath each of them. Parts that are amended in the revised manuscript are highlighted in blue in a manuscript file with marked revisions.

Response to Reviewer 2 Comments

Language can be further improved. For example, title: …, a comprehensive review; abstract: The molecular mechanism of the hepatitis E virus (HEV) pathology still is unclear. MicroRNAs (miRNAs), with the host or viral origin, interfere with virus replication and the host environment to create a suitable condition for producing mature HEV progeny. Understanding the biogenesis and interference of miRNAs with HEV will help reveal the mechanism of viral pathogenesis.

Response: As advised by the reviewer the language is improved for the entire manuscript.

Overlaps have been found in contexts between 1. Introduction and 2. Hepatitis E virus; it could be more concise.

Response: As recommended the overlaps between the introduction and 2nd paragraph were removed and in this context, the introduction is abridged.

Errors: 4.1. HEV-miR-A(2)6; 5.1(2). miRNA-214

Response: Thank you for this remark, errors in the title and numbering of the paragraph have been corrected

Round 2

Reviewer 1 Report

E. Golkocheva-Markova has provided a revised version of the review on the interplay between hepatitis E virus and micro RNAs (miRNAs). The manuscript has been improved and the author has addressed most of the points raised. A review on this topic is of great interest for the field.

 Minor comments:

Line 34: can cause chronic infection

Line 225: In the case of HCV

Line 345: conserved rather than conservative

Line 347: “This conservation can be the result of viral evolution in the context of adaptation to specific host environment” rather than “This conservatism can be assumed to be the result of viral evolution in contexts to the adaptation to specific host environment”

Line 591: suggested a possible link (rather than supposed)

Line 602-604: sentence difficult to understand.

Line 610: encodes

Line 623: are working

Line 623: delete “by”

Author Response

Dear Reviewer! Thank you for your time and valuable comments. 

Due to the recommendation, the following changes have been made:

Line 34: can cause chronic infection

Response: This was amended as recommended.

Line 225: In the case of HCV

Response: Full article added.

Line 345: conserved rather than conservative

Line 347: “This conservation can be the result of viral evolution in the context of adaptation to specific host environment” rather than “This conservatism can be assumed to be the result of viral evolution in contexts to the adaptation to specific host environment”

Response: Thank you, for lines 345 and 347 the words conservative and conservatism have been replaced

Line 591: suggested a possible link (rather than supposed)

Response: the supposed has been replaced by suggested

Line 602-604: sentence difficult to understand.

Response: In accordance with the reviewer’s comment on the sentence (lines 602-604) stylistic changes have been made to facilitate its understanding.

Line 610: encodes

Response: Thank you for this remark, on lines 610 the spelling mistake was corrected.

Line 623: are working

Line 623: delete “by”

Response: The suggestions for line 623 were done.